# Spectroscopy of a tunable moiré system with a correlated and topological flat band

Xiaomeng Liu [1,5], Cheng-Li Chiu[1,5], Jong Yeon Lee[2], Gelareh Farahi[1], Kenji Watanabe [3], Takashi Taniguchi [4], Ashvin Vishwanath[2] & Ali Yazdani [1✉]

Moiré superlattices created by the twisted stacking of two-dimensional crystals can host electronic bands with flat energy dispersion in which enhanced interactions promote correlated electron states. The twisted double bilayer graphene (TDBG), where two Bernal bilayer graphene are stacked with a twist angle, is such a moiré system with tunable flat bands. Here, we use gate-tuned scanning tunneling spectroscopy to directly demonstrate the tunability of the band structure of TDBG with an electric field and to show spectroscopic signatures of electronic correlations and topology for its flat band. Our spectroscopic experiments are in agreement with a continuum model of TDBG band structure and reveal signatures of a correlated insulator gap at partial filling of its isolated flat band. The topological properties of this flat band are probed with the application of a magnetic field, which leads to valley polarization and the splitting of Chern bands with a large effective g-factor.

[1] Joesph Henry Laboratories and Department of Physics, Princeton University, Princeton, NJ, USA. [2] Department of Physics, Harvard University, Cambridge, MA, USA. [3] Research Center for Functional Materials, National Institute for Materials Science, Tsukuba, Japan. [4] International Center for Materials Nanoarchitectonics, National Institute for Materials Science, Tsukuba, Japan. [5] These authors contributed equally: Xiaomeng Liu, Cheng-Li Chiu. ✉email: yazdani@princeton.edu

A moiré superlattice with flat electronic bands was first discovered by stacking two layers of graphene on top of each other at a precise angle[1–3]. The flat band in magic angle twisted bilayer graphene (MATBG) emerges from the interplay between interlayer hybridization and monolayer graphene' electronic structure and displays a plethora of correlated and topological phenomena. At partial band filling, MATBG shows a cascade of correlated insulating and superconducting phases, the underlying mechanisms of which are still being investigated[3–16]. A number of different Chern insulator phases, driven by either alignment of the MATBG with hexagonal BN or intrinsic strong correlations in such flat bands have also been found[17–21]. These discoveries motivate interest in other moiré systems and search for novel ways to create and control flat bands within them. A particularly promising moiré system is twisted double bilayer graphene (TDBG), in which the tunability of Bernal bilayer graphene band structure with an electric field can be exploited to create a tunable moiré system[22–26]. Previous theoretical and experimental studies have demonstrated that application of an electric field can isolate a flat band with a bandwidth of 10–20 meV[23,27–31] in TDBG. The electric field tunability makes it possible to realize this flat band in a wide range of twist angles (0.84°–1.53°) between the stacked bilayers, instead of at a specific angle as for the case of MATBG[23,24]. Corroborating this picture, previous transport experiments have shown that a robust correlated insulator appears in a confined electric field range at densities that correspond to half filling of a flat band in TDBG[22–26]. However, direct measurements of the band structure of this system and direct demonstration of its tunability, which underlies the interpretation of transport studies, has not been thus far performed.

We characterize the electronic structure of TDBG using scanning tunneling microscopy (STM) and spectroscopy as a function of carrier density and in the presence of a magnetic field. STM and gate-tuned scanning tunneling spectroscopy (GT-STS) have become a powerful tool to probe the properties of moiré systems, as its application to MATBG has provided critical information on the nature of flat bands in that system. STM experiments have not only confirmed the theoretical picture of the single particle band structure of MATBG but also have uncovered signatures of strong electronic correlation and topology at partial band filling of this system[7–10,19]. Applying these techniques to TDBG not only demonstrates its tunability but also provides spectroscopic signatures of correlation and topology of its flat bands that complements transport studies.

## Results and discussion

Figure 1a shows the schematic diagram of our device geometry, which we use to probe the electronic properties of TDBG samples, in a home built ultra-high vacuum STM at a temperature $T = 1.4$ K (see Methods for sample fabrication and STM measurement). Using STM imaging, we first obtain information on the structure and symmetry of the TDBG's moiré superlattice in our devices. Figure 1b shows an STM topography image, from which we can measure the periodicity and distortion of the moiré lattice to determine the twist angle of 1.48° and a very small heterostrain of 0.1% for this TDBG sample. Higher resolution STM images of TDBG samples (Fig. 1b inset) show the topmost graphene layer's triangular atomic lattice, which signifies the sensitivity of our experiments to the electronic properties of this layer (out of four) in this multilayer system. STM images also reveal the breaking of $C_2$ symmetry—a salient feature of TDBG, which is in contrast preserved in MATBG. This broken symmetry can be visualized by filtering the atomic lattice from STM images (Fig. 1b) and displaying the results in the moiré unit cell, as

shown in Fig. 1c. From the STM topographic height in such images, we identify three different locations, ABBC, ABCA, and ABAB, within the unit cell that correspond to different stacking configurations of the atoms between the layers in our sample[32]. While the locations corresponding to the AA stacking between the middle layers appear higher in STM images (Fig. 1b, c), there is also a measurable contrast in the height between AB or BA stacking locations as well. These locations are related by the $C_2$ symmetry of the moiré lattice, which is broken in TDBG. The broken $C_2$ symmetry is a distinguishing property of TDBG and is responsible for the gap between the conduction and the valence band at the Dirac points of its band structure, as well as the unique band topology of TDBG.

Spectroscopy and visualization of the electronic states of TDBG over a wide range of energies reveal the basic band structure of this system and validates the continuum model that has been used to understand its properties. Figure 1e shows the STS measurement at the charge neutral point (CNP) that displays four distinct peaks corresponding to the van Hove singularities (vHs) of the four lowest energy bands of this system. The excellent agreement between these measurements and the density of states calculated by the continuum model (Fig. 1f and Supplementary Material) allows us to attribute the four identified bands to the first conduction and valence bands: C1, V1, and the second conduction and valence band: C2, V2 (Fig. 1d). Further corroboration for this assignment can be found by comparing the spatial dependence of the electronic states measured using STM conductance maps at energies corresponding to vHs of these bands and the maps of the electron distribution on the top graphene layer for the corresponding bands from the continuum model (Supplementary Material). Figure 2 shows an example of this comparison illustrating that the points of high density of states for V2, C1, and C2 bands are located on the ABBC moiré sites, while V1 displays a different pattern with its maximum density of states on ABCA sites. There is not only good agreement between the experimental results and the continuum model but also such maps clearly visualize the broken $C_2$ symmetry in TDBG by displaying the difference between the density of states at ABAB and ABCA sites.

The tunability of TDBG's band structure can be demonstrated by the measurements of its spectroscopic properties as we vary the gate voltage of the silicon gate $V_{SiG}$ in our devices. In similar previous experiments on gated devices with STM, such as those on MATBG (for example ref. [11]), varying $V_{SiG}$ allowed us to study the properties of such two-dimensional systems as a function of carrier concentration. However, for studies of TDBG, it is important to recognized $V_{SiG}$ tunes both the carrier concentration n and the electric field D simultaneously: $n = CV_{SiG}/e + n_0$, $D = CV_{SiG}/2 + D_0$, where $C$ is the capacitance per area between TDBG and the silicon gate, $n_0$ and $D_0$ are the intrinsic doping and the built-in electric field. Measurements of the tunneling spectra display a rich array of features that shift systematically as a function of $V_{SiG}$, as shown in Fig. 3a. First, we highlight that changes of doping in TDBG caused by the $V_{SiG}$ results in three clear jumps of the vHs marked by dashed white lines, indicating the Fermi energy is passing through a band gap. From these jumps, we identify the gate voltages corresponding to CNP ($V_{SiG} = -3.5$ V), full occupancy of the conduction band (72.5 V), and full occupancy of the valence band (−79.5 V). The appearance of CNP near zero gate voltage reveals that samples are not doped by impurities or by a significant work function difference between the sample and the STM tip. Furthermore, we find the gate voltage ranges required to occupy the conduction band and the valance band are identical: $\Delta V_{SiG} = 76$ V, which is also consistent with the carrier density needed to fill a moiré band of the 1.48° TDBG: $n_S = 5.08*10^{12}$ cm$^{-2}$ based on a parallel plate capacitance

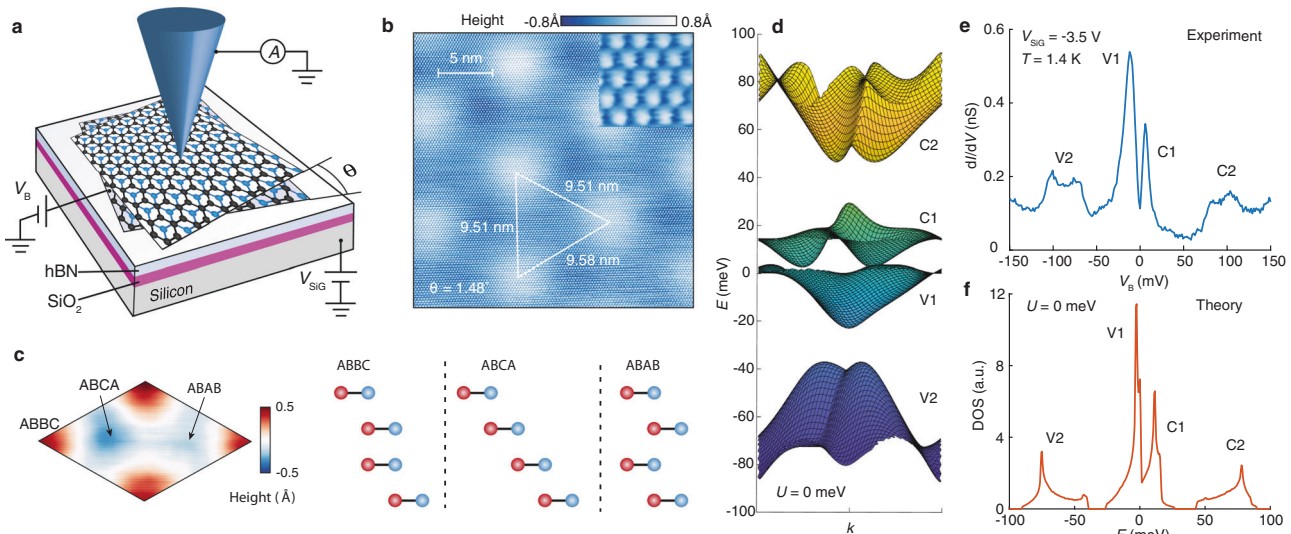

**Fig. 1 STM on TDBG. a** Schematic of the STM measurement setup on TDBG devices. **b** STM topograph of TDBG, obtained with bias setpoint $V_{set} = -400$ mV, current setpoint $I_{set} = 20$ pA and gate voltage $V_{SiG} = -88$ V. Top right inset shows magnified image (1 nm size) with top graphene layer lattice in a Bernal bilayer. The bright spots in the inset are carbon atoms. **c** Filtered topograph removing atomic lattices to emphasize different stacking orders of TDBG plotted in the moiré unit cell. The colored circles on the right represent carbon atoms on A and B sublattices of each layer. **d** Calculated band structure of TDBG with a twist angle of $\theta = 1.48°$ under zero potential different between the top and the bottom graphene layers ($U = 0$). **e** Tunneling spectrum of 1.48° TDBG device at charge neutrality, measured with $V_{set} = -400$ mV, $I_{set} = 400$ pA and AC modulation of $V_{mod} = 3$ mV. **f** Density of state on the top graphene layer calculated by a continuum model under zero electric field.

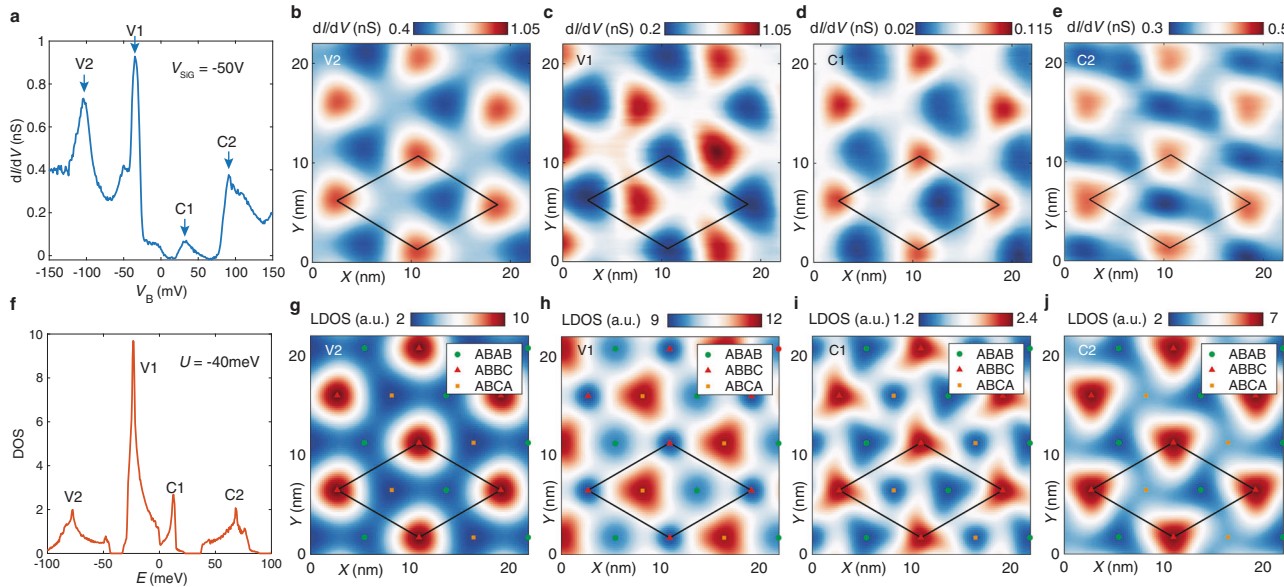

**Fig. 2 Visualizing Electronic States of TDBG. a** Tunneling spectrum at $V_{SiG} = -50$ V and $T = 1.4$ K, measured with $V_{mod} = 5$ mV, $V_{set} = -200$ mV, $I_{set} = 400$ pA. **f** Calculated DOS with parameter $U = -40$ meV, which is the onsite potential difference between the top and bottom graphene layer. **b–e** Conductance maps for V2, V1, C1 and C2 bands, taken at $T = 1.4$ K, $V_{SiG} = -50$ V, $V_{mod} = 5$ mV. The voltage setpoint of each map is set to the spectrum peaks in **a** at $V_{set} = -103$, $-35$, $32$, and $91$ mV, while the current setpoint is chosen to keep the tip at roughly the same height as **a**. The plotted data was smoothed with a 0.8 nm radius filter. The diamond shapes represent a moiré unit cell. **g–j** Calculated electron density distribution on the top graphene layer for the four bands. $U = -40$ meV is used for this calculation. Shown electron densities are summed over each band (Supplementary Material), which appears to capture the data under these STM setup conditions. The diamonds match with the ones in **b–e**. Different stacking orders are shown by different colored symbols.

model (see Methods). These observations indicate that our sample is pristine, and our measurements are free from artifacts of tip-induced band bending, which was a concern in early STM studies of MATBG[7–10].

The structure of TDBG bands, especially those of the C1 and the V1 bands, undergoes drastic changes in spectroscopic measurements as we tune the electric field $D$ in our device by changing $V_{SiG}$. Before increasing $D$, at the CNP, there are large band gaps between V2 and V1 ($\Delta_{-n_s}$), C1 and C2 ($\Delta_{n_s}$), with $\Delta_{-n_s} < \Delta_{n_s}$ (Fig. 1e). Figure 3 (a, c, and d) show that increasing the magnitude of the electric field $|D|$ with increasing $|V_{SiG}|$ results in reduction of the magnitudes of these energy gap, with $\Delta_{-n_s}$ being

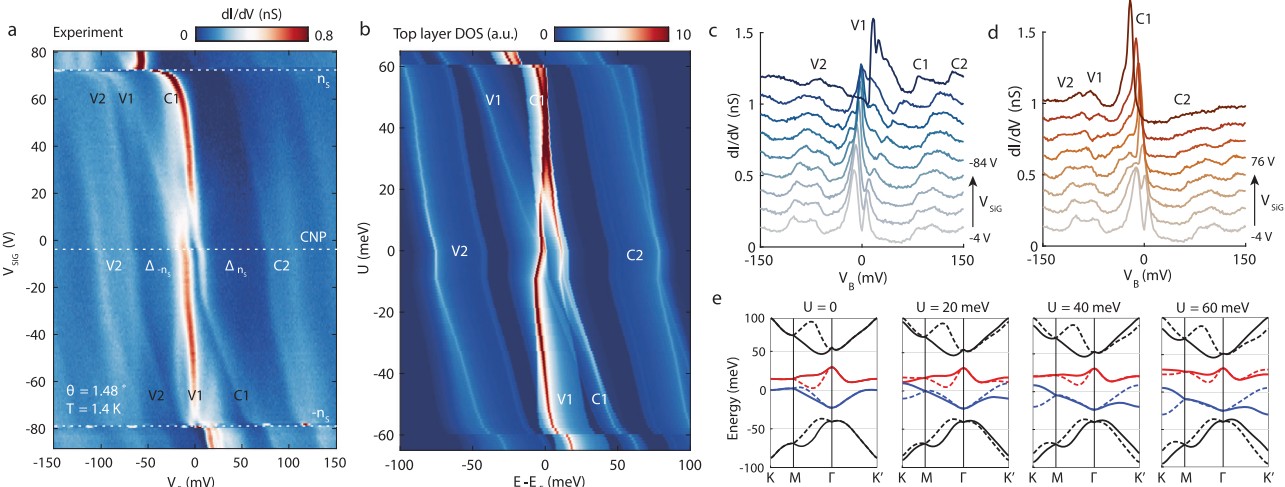

**Fig. 3 Tunable band structure of TDBG. a** Differential tunneling conductance as a function of bias voltage $V_B$ and silicon gate $V_{SiG}$ measured near the ABAB stacking configuration, with $V_{set} = -400$ mV, $I_{set} = 400$ pA and AC modulation of $V_{mod} = 3$ mV. The white dashed lines mark the charge neutral point (CNP) and full filling of C1 ($n_S$) and V1 (-$n_S$) bands. **b** Calculated top graphene layer density of state as a function of energy respect to Fermi energy $E - E_F$ and onsite potential difference between the top and bottom graphene layer $U$. A different bias range is used in panel b (±100 meV, instead of ±150 meV in panel **a**) to make the best visual comparison with panel **a**. This energy mismatch with experimental features could be caused by electron interactions, which are not accounted for in the theory or inaccurate model parameters. **c**, **d** line spectra from CNP to full hole (**c**) and electron (**d**) filling. **e** Band structure at different $U$ in the mini-Brillouin zone, red (blue) lines represent the first conduction (valence band). The solid and dashed lines correspond to two different valleys $K_1$ and $K_2$ of the Brillouin zone of the graphene lattice.

almost eliminated by large electric fields, as displayed at the top and bottom portions of Fig. 3a. These gaps reduce in a symmetric manner toward the positive and negative gate voltages, signifying that the built-in electric $D_0$ is close to zero. These findings are consistent with previous transport studies of TDBG and show that the band structure of TDBG can be tuned with an electric field. In addition to these gaps, we identify a smaller gap between C1 and V1 bands at charge neutrality $\Delta_{CNP}$, which behaves in an asymmetric fashion with respect to the sign of the $V_{SiG}$. At positive values of $V_{SiG}$, $\Delta_{CNP}$ is first reduced in magnitude to zero but it reopens again with increasing $V_{SiG}$ (Fig. 3a, d), in agreement with transport experiments at this twist angle[23]. In contrast, $\Delta_{CNP}$ monotonously increases in magnitude as $V_{SiG}$ is made more negative (Fig. 3a, c). This asymmetry with respect to $V_{SiG}$ is due to the polarizing effect of the electric field, which induces changes in the spatial distribution of electronic states across the different atomic layers of TDBG. Our experiments are sensitive to this asymmetry because they are sensitive to the electronic properties of the topmost graphene layer in TDBG. In fact, comparing our experimental results of the gate-tuned measurements of tunneling spectra to the continuum model (Supplementary Material), we find the calculated density of states for the top graphene layer (Fig. 3b) from this model shows a remarkable resemblance to our data (Fig. 3a), including the observed asymmetric behavior with respect to $V_{SiG}$. In these calculations, the tuning of electric field $D$ is modeled by the adjustable parameter $U$, which is the onsite potential difference between the top and bottom graphene layer, spread across the four graphene layers (Supplementary Material). Furthermore, the continuum model allows us to understand why we do not observe the closing of $\Delta_{CNP}$ at negative $V_{SiG}$—it actually closes, but the signature appears only in the density of states of the bottom layer, which is not being probed by our experiment (Supplementary Fig. 1). One might notice a slight discrepancy between the model and the experiment, that there is a small overlap between V1 and C1 in the calculated band structure (Figs. 1d and 3b), in contrast to the clear CNP gap seen in our experiment and transport experiment[23]. In fact, with our choice of model parameters, a CNP gap at $D = 0$ and its closing

and opening with the application of $D$ is predicted for a twist angle larger than 1.52°. This minor discrepancy can thus be explained by electron interactions, coupling with the substrate or inaccurate model parameters. Lastly, we find that the bandwidth of C1 in both experiment and model calculation is smaller than that of V1, which explains why correlated insulators have only been discovered in the conduction band and establishes the setting for the correlation signatures to emerge in our experiments.

The influence of strong electronic correlation in the TDBG system can be examined in higher resolution gate-tuned spectroscopy measurements of the C1 and V1 bands. Figure 4a shows such measurements in which we find a number of different features as the occupation of the C1 band is tuned. First, we note the abrupt jump in vHs of C1 at CNP due to the CNP gap, the closing of which can be identified near $V_{SiG} \sim 20$ V in this higher resolution measurement. Second, as the C1 is being occupied, we also observe a sudden increase in the broadening of the vHs of the fully occupied V1 band, reminiscent of similar features seen in such measurements of MATBG and understood as a results of strong correlation in that system[8]. Third and most intriguingly, increasing the occupation of C1 to half filling (at $V_{SiG} = 34.5$ V), where transport studies have uncovered a correlated insulator, we observe a sudden jump of the vHs of this band (Fig. 4a–c and Supplementary Figs. 2 and 3). This jump sometimes appears together with a suppression of the density of states at the Fermi level, the measurements of which we find to be sensitive to the location of the tip and also likely to its work function (inset Fig. 4a and discussion on Figs. S2 and S3 in Methods). Examining the behavior of vHs more closely by tracking its location in energy at different $V_{SiG}$, we extract the magnitude of its jumps $\Delta$ at different twist angle areas. Consistent with correlation gaps extracted from transport studies at half filling of the C1 band, we find the magnitude of $\Delta$ increases from 0.38 to 1.3 meV as the twist angle decreases from 1.48° to 1.43°. The observation of jump-like features in our data is not expected from a simple insulating gap, such as observed single particle gaps observed at CNP and full occupancy of C1 and V1 bands. Its shape is however reminiscent of similar measurements on MATBG near every

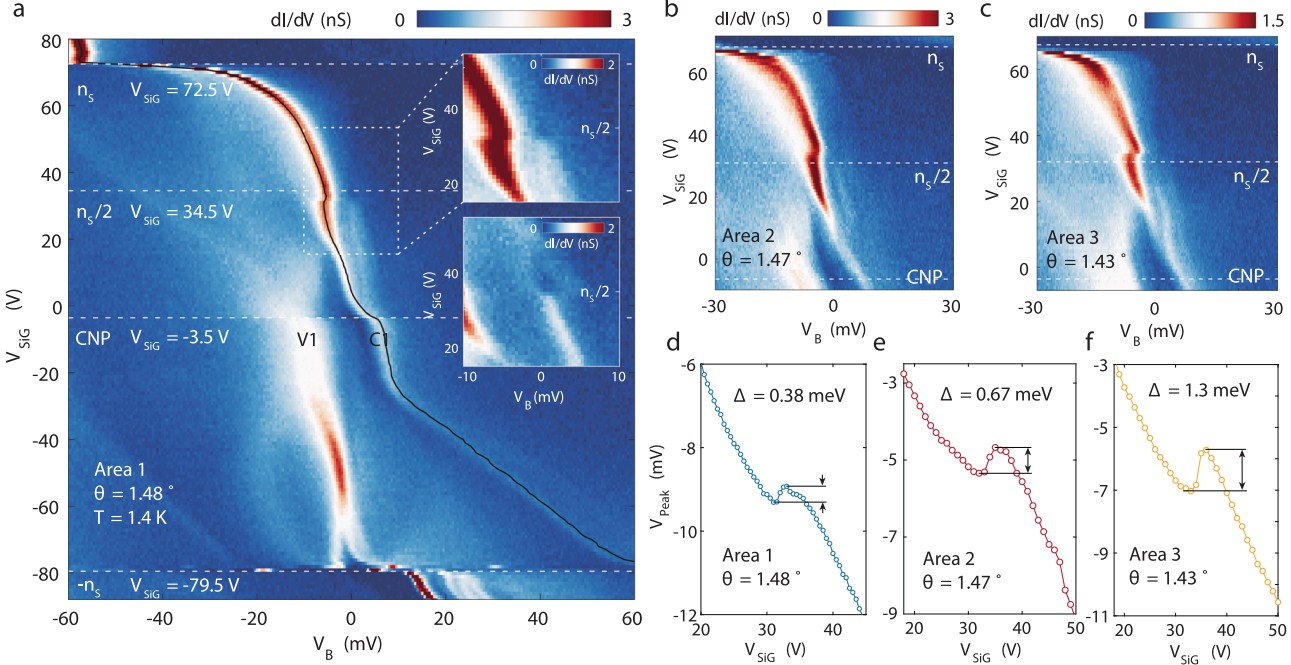

**Fig. 4 Evidence for correlated insulating states at half filling of the conduction band. a** High-resolution gate-dependent spectra in the first area with $\theta = $ 1.48°, taken near the ABAB stacking configuration with $V_{set} = -400$ mV, $I_{set} = 800$ pA and $V_{mod} = 0.5$ mV. There is a clear jump of vHs near the half filling of the conduction band. Top inset, zoom-in around the half filling and zero bias, revealing a small dip in tunneling conductance at zero bias near half filling. Bottom inset, a zoom-in of Supplementary Fig. 2, showing a more pronounced gap near half filling. **b, c** High-resolution maps showing similar jumps of vHs in different areas of the sample with twist angles 1.47° and 1.43°, also taken with $V_{mod} = 0.5$ mV. **d–f** Traces of vHs peaks over silicon gate voltages in three different areas. The magnitude of the jump $\triangle$ in each area is extracted.

quarter fillings of that system[11,12] and requires further measurements and theoretical analysis.

Lastly, we show that topology of the band structure of the TDBG band can be revealed by examining its spectroscopic properties in a magnetic field. In the presence of an electric field $D$, the C1 band of TDBG in our theoretical model carries a topological valley Chern number $C_v = 2$, meaning Chern number $C = 2$ in one valley and $C = -2$ in the opposite valley. Since a Chern band generates orbital magnetic moments, it would couple strongly to a perpendicular magnetic field. In Fig. 5a, we show a gate-dependent STS measurement under a perpendicular magnetic field of 4 T. Compared with zero field data (Fig. 4a), the most salient feature is the splitting of the vHs of the C1 band. This splitting persists over a large range of densities and occur below Fermi level, thus is likely due to single particle band effect. This is in contrast to the splitting induced by interactions, which occurs on the Fermi energy near half filling. The behavior of the splitting broadly agrees with the expectation of a topological valley Chern band. Since the two valleys of the C1 band carry opposite Chern numbers and orbital g-factors, they shift in energy under a magnetic field in the opposite directions by $\pm g\mu_B B$, where $\mu_B$ is the Bohr magneton and $g$ is the magnitude of the orbital g-factor (Fig. 5d, e). While in general g-factors depend on crystal momentum, here for simplicity we discuss the average g-factor[28]. Thus, the magnetic field split the valley-degenerate vHs of the C1 band under zero magnetic field into two vHs of different valleys separated by $2g\mu_B B$. Figure 5b and Supplementary Fig. 4 shows the evolution of C1 splitting as a function of perpendicular magnetic fields. Although the energy of the C1 peaks move irregularly between different fields, possibly caused by the tip position change when changing the magnetic field, the separation between the two peaks evolves consistently. Extracting this separation, we find they collapse onto lines with very large g-factors of $g = 13$ and 21 for half filling and three-quarter filling (Fig. 5c). The large

g-factor rules out the possibility that the splitting is caused by spin Zeeman effect, since transport studies under in-plane magnetic field consistently find spin g-factor close to two[22–26]. From the continuum model calculation, orbital g-factor averaged in the mini-Brillion zone is between 6 and 7 for most $U$ (Supplementary Material), which is a factor of two times smaller than the experimental value. However, the orbital g-factor calculation sensitively depends on the energy landscape within the narrow C1 band, which may be altered by interactions that are not accounted for in the single particle picture. Without knowing the exact shape of the band, a simple picture below may provide a rough estimate of the g-factor amplitude for a general $C_v = 2$ band. Since the CNP gap possesses valley Chern numbers, at the $K_1$ ($K_2$) valley, the CNP gap would shift in energy from the bottom (top) edge of C1 (V1) band at zero magnetic field to the top (bottom) of C1 (V1) band at half flux quantum per moire unit cell[33] (Fig. 5e). If the vHs of C1 roughly shift along with the CNP gap, and if we assume the CNP gap is much smaller than the bandwidth of C1 and V1, the splitting between vHs of the C1 band in two valleys produced by the field would be on the order of $B/B_{1/2\Phi}*(w_c + w_v)$, where $B_{1/2\Phi} = 26.3$ T is the magnetic field of half flux quantum per moire unit cell and $w_c$ ($w_v$) is the bandwidth of the C1 (V1) band. Assuming the bandwidth of C1 and V1 are on the order of 20 meV, this formula produces a g-factor $g = 13$, roughly agreeing with our observation. Beside the splitting, we also resolve a discontinuity of the vHs just below $V_{SiG} = 20$ V, where the CNP gap collapses in Fig. 4a, and a jump of vHs near three-quarter filling of the C1 band. These features and other topological signatures of this system that are revealed in the present of a magnetic field[33], can be explored in future studies of this system.

## Methods
**Sample preparation.** TDBG samples are stacked on a hexagonal Boron Nitride (hBN) layer on SiO₂/silicon substrate and electrically contacted with gold electrodes.

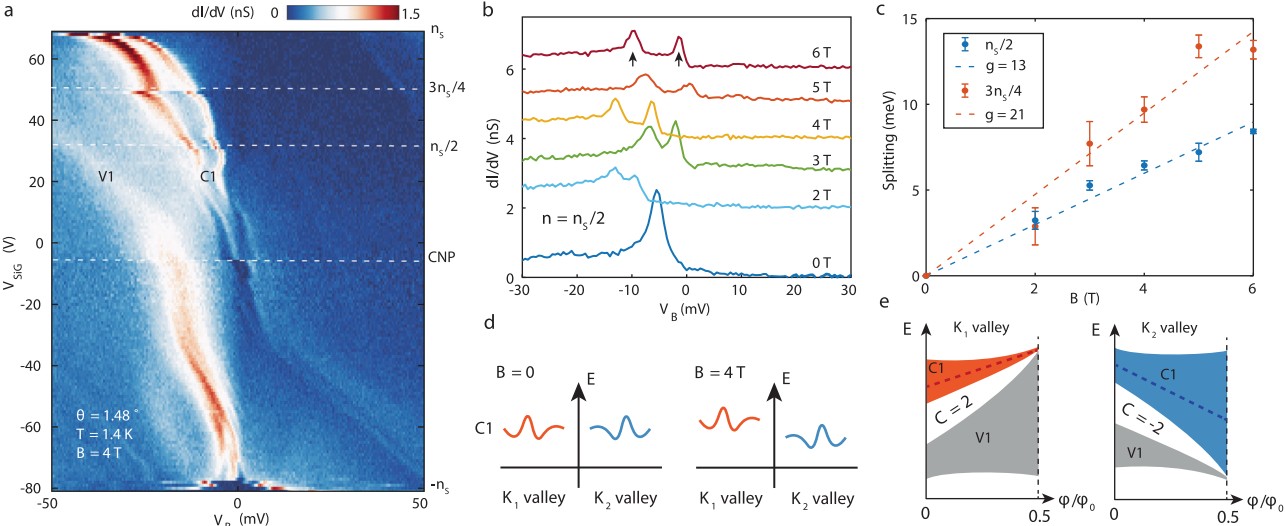

**Fig. 5 Splitting of vHs under perpendicular magnetic fields. a** STS measurements between $-n_S$ and $n_S$ under a perpendicular magnetic field of 4 T, taken with $V_{set} = -400$ mV, $I_{set} = 800$ pA and $V_{mod} = 0.5$ mV. **b** Tunneling spectra at half filling of the C1 band under various perpendicular magnetic fields. Traces are offset for clarity. **c** Splitting of the C1 vHs as a function of perpendicular magnetic field $B$ at half filling (blue symbols) and three-quarter filling (red symbols), obtained from Gaussian fitting of the split peaks. The error bars represent 95% confidence bounds of the fit. The dashed lines are the linear fit of the splitting, corresponding to $\Delta E = 2g\mu_B B$ with g-factor noted in the legend. **d** schematic of C1 band's response to magnetic fields. The two valleys (red and blue) shift in opposite energy directions due to their opposite valley Chern numbers. **e** illustration of valley splitting estimation based on Chern number of the band. On $K_1$ ($K_2$) valley, the CNP gap between V1 and C1 carries Chern number $C = 2$ ($C = -2$), which move to the top of C1 band (bottom of V1 band) at half flux per moiré unit cell. The red and blue dashed lines represent the vHs of C1 band in the two valleys, and $w_c$ and $w_v$ denote the bandwidth of C1 and V1 band at zero field.

The samples are fabricated by tearing and stacking an exfoliated Bernal stacked bilayer graphene. We fabricated our samples in similar ways to published STM results on MATBG[8,11]. Our pickup stamp is made from a polyvinyl alcohol (PVA) coated transparent tape, which cover a polydimethylsiloxane (PDMS) block on a glass slide handle. Using the PVA stamp, we first pick up an exfoliated hBN flake. We then use the hBN to tear and pickup half of an exfoliated Bernal bilayer graphene. After rotating the stage about 1.3°, we pick up the remaining bilayer graphene flake. To flip the stack upside down, we transfer the stack onto a new PDMS block and dissolve PVA with water to detach the original pickup stamp. After cleaning in water, we drop the flipped stack from the PDMS block to a SiO₂/Si substrate with prepatterned gold contact. Once the device is made, we briefly dip it into water to clean PVA residues and then annealed it in UHV at 180 °C overnight.

**STM measurements**. The experiment is done in a home built UHV STM with (6,1,1) vector magnet operating at $T = 1.4$ K. The measurements are performed with a tungsten tip prepared and characterized on a Cu(111) single crystal. Through controlled indentation, we shape the tip until its poke mark is confined and its spectrum featuring the Cu(111) surface state at the right energy. We then locate the TDBG sample with the capacitance guiding technique[34]. The topography is taken under constant current mode, and the differential conductance is acquired by standard lock-in method with AC modulation at 4 kHz.

All spectrums presented in this study with the exception of Fig. 2a are measured with a freshly prepared tip on the first landing spot before any scanning. We note that the current generation of samples have some residue contamination, which sometimes modifies the spectrums after scanning to obtain images of the sample. Supplementary Fig. 5a is spectrum measured with a freshly prepared tip on the first landing spot on the sample. After scanning, same measurement was repeated on the same spot, which yield the spectrum shown in panel b. Comparing the two, the general four band features survive scanning. However, the vHs of v1 and c1 appear further apart and both vHs drift further away from the Fermi energy. To demonstrate this is not just random tip-to-tip variation and fresh prepared tip yield reproducible result, we show a spectrum measured with a different fresh prepared tip on a different part of the sample in panel c of Supplementary Fig. 5. Although the twist angle is slightly different, the features in panel c is much closer to the panel a, both measured with freshly tip and are distinctly different from panel b.

**Estimated silicon gate voltage required to fill a moiré band**. The moiré unit cell area for our 1.48° TDBG is $A = 78.7$ nm². Thus, the electron density corresponds to full filling is $n_S = 4/A = 5.08*10^{12}$ cm⁻². In our experiment, the SiO₂ thickness is ~285 nm and the hBN thickness is estimated to be ~30 nm. Assuming the dielectric constants of SiO₂ and hBN are both $\epsilon_r = 4$, the gate voltage need to dope the TDBG from CNP to $n_S$ is then $\Delta V_{SiG} = 72.4$V, similar to the experimental value of 76 V.

**Estimate of displacement field of hall-filled state**. According to $D = CV_{SiG}/2 + D_0$, in order to calculate the displacement field at half filling, we need to know the build-in electric field $D_0$. We can estimate $D_0$ by examining the gap between C1 and V1 vHs as a function of gate voltage (Supplementary Fig. 6). Assuming the gap between C1 and V1 vHs is symmetric over positive and negative displacement field (which is verified by the continuum model), where the two lines in Supplementary Fig. 6 intercept ($V_{SiG} = -10$V) corresponds to zero displacement field. We can then derive the displacement field at half filling to be around $(34.5 \text{ V} - (-10 \text{ V}))/300 \text{ nm}*4/2 \sim 0.30$ V/nm, where 34.5 V is the gate voltage of half filling, 300 nm is the dielectric thickness and 4 is the dielectric constant of SiO₂ and hBN. This is close to the lower bound of the electric field where transport study observes correlated insulator states (lower bound of 0.28–0.37 V/nm is reported from two devices with nearby twist angle in ref. [23]).

**Estimate of tip gating effect**. Applying bias voltage could potentially create additional doping and displacement field from the tip. However, the estimate here shows the tip gating effect should be negligible considering the bias and gate voltage range (300 mV and 165 V, respectively) explored in our study. To estimate the tip gating effect, we zoom-in near the full filling of the valance band (Supplementary Fig. 7). We observe clear jumps of spectral features when the Fermi energy crosses the full filling gaps. In the case that the tip does not have any gating effect, these jumps would occur at the same back gate voltage independent on bias voltage. However, tracing the full filling jumps, we notice a small change of fulling filling gate voltage depending on the bias voltage (Supplementary Fig. 7). From the slope of the discontinuity line, we can estimate the gating efficiency ratio between the tip and the back gate, which find 300 mV of the tip bias gating is comparable to 1 V of back gate voltage.

**Details on half-filling states**. One may notice the half-filled states in Fig. 4 does not occur exactly at half filling. While we do not know the exact cause of this, it could be a result of nonlinear gating effect caused by mobile charged impurities inside the hBN or tip-induced band-bending effect[8]. We also cannot rule out the possibility that the correlated insulating state occurs slightly away from half filling in large twist angle TDBG devices. It is worth noting in previous transport studies, although the correlated insulator is commonly observed at half filling, sometimes it occur a little off from exact half filling[23].

**Discussion on Supplementary Figs. 2 and 3**. In Fig. 4a–c and Supplementary Figs. 2 and 3, we show several GT-STS measurements that provide information on the variation of such measurements at different areas of the same device as well as different locations within the moiré unit cell. Supplementary Fig. 2 shows a different type of data showing the signatures of correlations in the broadening and the visibility of the gap at the Fermi level in C1 can be different when measured on

different locations or with different tip conditions. In Supplementary Fig. 3, we include more GT-STS measurements presented together with STM topographs that show areas 1–4. The location of the corresponding measurements within the unit cell are marked on these topographs (Supplementary Fig. 3a–d). Spectrums in Supplementary Fig. 2 (Supplementary Fig. 3h), which shows two peaks of C1 band, with one peak crossing the Fermi level and further split around the half filling, appear to the very different from other data set (Supplementary Fig. 3a–c). The difference could be caused by the different locations where the data is taken, but it could also be caused by anomalous tip condition in acquiring Supplementary Fig. 2.

## Data availability

The data that support the findings of this study are available from the corresponding author upon reasonable request.

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

## Acknowledgements

We thank Andrei Bernevig for helpful discussions. A.V., J.Y.L., and X.L. would like to thank Zeyu Hao, Eslam Khalaf, Shang Liu, and Philip Kim for an earlier collaboration on TDBG. This work was primarily supported by the Gordon and Betty Moore Foundation's EPiQS initiative grants GBMF4530, GBMF9469, and DOE-BES grant DE-FG02-07ER46419 to A.Y. Other support for the experimental work was provided by NSF-MRSEC through the Princeton Center for Complex Materials NSF-DMR-1420541, NSF-DMR-1904442, NSF-DMR-2011750, ExxonMobil through the Andlinger Center for Energy and the Environment at Princeton, and the Princeton Catalysis Initiative. A.V. and J.Y.L were supported by a Simons Investigator fellowship. K.W. and T.T. acknowledge support from the Elemental Strategy Initiative conducted by the MEXT, Japan, grant JPMXP0112101001, JSPS KAKENHI grant JP20H00354, and the CREST (JPMJCR15F3), JST.

## Author contributions

X.L., C.C., and A.Y. designed the experiment. X.L. and C.C. fabricated the samples. X.L., C.C., and G.F. performed the STM measurements and analyzed the data. J.Y.L., A.V., and X.L. conducted the theoretical calculations. K.W. and T.T. provided hBN crystals. X.L., C.C., J.Y.L., and A.Y. wrote the paper with input from all authors.

## Competing interests

The authors declare no competing interests.
