## [Peer Review File · Nature Communications]

REVIEWER COMMENTS

Reviewer #1 (Remarks to the Author):

The authors have improved their manuscript following my comments and those of the other reviewers. I am happy to recommend publication of this work in Nature Communications.

Reviewer #2 (Remarks to the Author):

The authors have revised their manuscript and responded to most concerns of the reviewers of the first round. However there are still some points to address before the manuscript can be published.

In his first comment, reviewer #1 pointed the difference in energy scale between the figures 1e and 1f which is as already stated confusing for the reader especially since the agreement with experiments is qualified as excellent in the manuscript. This should be commented in the manuscript.

The technical issue described in the reply to the fifth comment of reviewer #2 raises suspicion about the experimental results and deserves clarification. How robust are the spectroscopic measurements to STM scanning? In their response the authors say that "the more subtle features will be altered or destroyed" presumably due to some molecules at the surface. They don't specify which features are altered or destroyed. Are they talking about the splitting of the resonances and so on? I would guess no but a comparison of the spectrum before and after scanning at the same place would completely clarify this issue.

Reviewer #3 (Remarks to the Author):

The authors have responded satisfactorily to the reviewers' reports. The manuscript is appropriate in Nature Communications.

We thank the referees for reviewing our manuscript for the second time. We are glad that referee #1 and #3 have recommended our manuscript to be published. Here we address referee #2's comments:

Referee #2 (Remarks to the Author):

The authors have revised their manuscript and responded to most concerns of the reviewers of the first round. However there are still some points to address before the manuscript can be published.

In his first comment, reviewer #1 pointed the difference in energy scale between the figures 1e and 1f which is as already stated confusing for the reader especially since the agreement with experiments is qualified as excellent in the manuscript. This should be commented in the manuscript.

Response: We agree with the referee's comment and added the following passage in revised caption of Fig. 1: "A different bias range is used in panel b ($\pm 100\text{meV}$, instead of $\pm 150\text{meV}$ in panel a) to make the best visual comparison with panel a. This energy mismatch with experimental features could be caused by electron interactions which are not accounted for in the theory or inaccurate model parameters."

The technical issue described in the reply to the fifth comment of reviewer #2 raises suspicion about the experimental results and deserves clarification. How robust are the spectroscopic measurements to STM scanning? In their response the authors say that "the more subtle features will be altered or destroyed" presumably due to some molecules at the surface. They don't specify which features are altered or destroyed. Are they talking about the splitting of the resonances and so on? I would guess no but a comparison of the spectrum before and after scanning at the same place would completely clarify this issue.

Response:

To clarify, we note that the current generation of samples have some residue contamination which sometimes modifies the measurements after scanning to obtain images of the sample. We have found that measurements with fresh prepared tip that lands on the sample, prior to scanning to be highly reproducible. Therefore, the spectra presented in the paper are obtained under this condition (which we note in methods section STM measurements). Following the referee's suggestion, we show below examples of measurements that clarifies this point below: In Fig. R1, panel a is spectrum measured with a fresh prepared tip on the first landing spot on the sample. After scanning, same measurement was repeated on the same spot, which yield the spectrum shown in panel b. Comparing the two, the general four band features survive scanning. However, the vHs of v1 and c1 appear further apart and both vHs drift further away from the Fermi energy. To demonstrate this is not just random tip-to-tip variation, we show a spectrum measured with a different fresh prepared tip on a different part of the sample in panel c of Fig. R1. Although the twist angle is slightly different, the features in panel c is much closer to the panel a, both measured with fresh tips and are

distinctly different from panel b. We have included these data in Fig. S5 of the updated manuscript with a brief discussion in methods.

Figure R1. Effect of scanning on measured tunneling spectrum. Panel a and c show spectrums measured with fresh prepared tips without scanning. Panel b shows spectrum measured at the same spot as panel a after scanning images with the tip in panel a.